# Learning Value Functions in Deep Policy Gradients using Residual Variance

**Yannis Flet-Berliac**[*]
Inria, Scool team
Univ. Lille, CRIStAL, CNRS
`yannis.flet-berliac@inria.fr`

**Reda Ouhamma**[*]
Inria, Scool team
Univ. Lille, CRIStAL, CNRS
`reda.ouhamma@inria.fr`

**Odalric-Ambrym Maillard**
Inria, Scool team

**Philippe Preux**
Inria, Scool team
Univ. Lille, CRIStAL, CNRS

## Abstract

Policy gradient algorithms have proven to be successful in diverse decision making and control tasks. However, these methods suffer from high sample complexity and instability issues. In this paper, we address these challenges by providing a different approach for training the critic in the actor-critic framework. Our work builds on recent studies indicating that traditional actor-critic algorithms do not succeed in fitting the true value function, calling for the need to identify a better objective for the critic. In our method, the critic uses a new state-value (resp. state-action-value) function approximation that learns the value of the states (resp. state-action pairs) relative to their mean value rather than the absolute value as in conventional actor-critic. We prove the theoretical consistency of the new gradient estimator and observe dramatic empirical improvement across a variety of continuous control tasks and algorithms. Furthermore, we validate our method in tasks with sparse rewards, where we provide experimental evidence and theoretical insights.

## 1 Introduction

Model-free deep reinforcement learning (RL) has been successfully used in a wide range of problem domains, ranging from teaching computers to control robots to playing sophisticated strategy games (Silver et al., 2014; Schulman et al., 2016; Lillicrap et al., 2016; Mnih et al., 2016). State-of-the-art policy gradient algorithms currently combine ingenious learning schemes with neural networks as function approximators in the so-called actor-critic framework (Sutton et al., 2000; Schulman et al., 2017; Haarnoja et al., 2018). While such methods demonstrate great performance in continuous control tasks, several discrepancies persist between what motivates the conceptual framework of these algorithms and what is implemented in practice to obtain maximum gains.

For instance, research aimed at improving the learning of value functions often restricts the class of function approximators through different assumptions, then propose a critic formulation that allows for a more stable policy gradient. However, new studies (Tucker et al., 2018; Ilyas et al., 2020) indicate that state-of-the-art policy gradient methods (Schulman et al., 2015; 2017) fail to fit the true value function and that recently proposed state-action-dependent baselines (Gu et al., 2016; Liu et al., 2018; Wu et al., 2018) do not reduce gradient variance more than state-dependent ones.

These findings leave the reader skeptical about actor-critic algorithms, suggesting that recent research tends to improve performance by introducing a bias rather than stabilizing the learning. Consequently, attempting to find a better baseline is questionable, as critics would typically fail to fit it (Ilyas et al., 2020). In Tucker et al. (2018), the authors argue that "much larger gains could be achieved by instead improving the accuracy of the value function". Following this line of thought, we are interested in ways to better approximate the value function. One approach addressing this issue is to put more focus on relative state-action values, an idea introduced in the literature on advantage reinforcement

---

[*]Equal contribution.

learning (Harmon & Baird III) followed by works on dueling (Wang et al., 2016) neural networks. More recent work (Lin & Zhou, 2020) also suggests that considering the *relative action values*, or more precisely the ranking of actions in a state leads to better policies. The main argument behind this intuition is that it suffices to identify the optimal actions to solve a task. We extend this principle of relative action value with respect to the mean value to cover both state and state-action-value functions with a new objective for the critic: minimizing the variance of residual errors.

In essence, this modified loss function puts more focus on the values of states (resp. state-actions) relative to their mean value rather than their absolute values, with the intuition that solving a task corresponds to identifying the optimal action(s) rather than estimating the exact value of each state. In summary, this paper:

- Introduces **A**ctor with **V**ariance **E**stimated **C**ritic (AVEC), an actor-critic method providing a new training objective for the critic based on the residual variance.

- Provides evidence for the improvement of the value function approximation as well as theoretical consistency of the modified gradient estimator.

- Demonstrates experimentally that AVEC, when coupled with state-of-the-art policy gradient algorithms, yields a significant performance boost on a set of challenging tasks, including environments with sparse rewards.

- Provides empirical evidence supporting a better fit of the true value function and a substantial stabilization of the gradient.

## 2    RELATED WORK

Our approach builds on three lines of research, of which we give a quick overview: policy gradient algorithms, regularization in policy gradient methods, and exploration in RL.

Policy gradient methods use stochastic gradient ascent to compute a policy gradient estimator. This was originally formulated as the REINFORCE algorithm (Williams, 1992). Kakade & Langford (2002) later created conservative policy iteration and provided lower bounds for the minimum objective improvement. Peters et al. (2010) replaced regularization by a trust region constraint to stabilize training. In addition, extensive research investigated methods to improve the stability of gradient updates, and although it is possible to obtain an unbiased estimate of the policy gradient from empirical trajectories, the corresponding variance can be extremely high. To improve stability, Weaver & Tao (2001) show that subtracting a baseline (Williams, 1992) from the value function in the policy gradient can be very beneficial in reducing variance without damaging the bias. However, in practice, these modifications on the actor-critic framework usually result in improved performance without a significant variance reduction (Tucker et al., 2018; Ilyas et al., 2020). Currently, one of the most dominant on-policy methods are proximal policy optimization (PPO) (Schulman et al., 2017) and trust region policy optimization (TRPO) (Schulman et al., 2015), both of which require new samples to be collected for each gradient step. Another direction of research that overcomes this limitation is off-policy algorithms, which therefore benefit from all sample transitions; soft actor-critic (SAC) (Haarnoja et al., 2018) is one such approach achieving state-of-the-art performance.

Several works also investigate regularization effects on the policy gradient (Jaderberg et al., 2016; Namkoong & Duchi, 2017; Kartal et al., 2019; Flet-Berliac & Preux, 2019; 2020); it is often used to shift the bias-variance trade-off towards reducing the variance while introducing a small bias. In RL, regularization is often used to encourage exploration and takes the form of an entropy term (Williams & Peng, 1991; Schulman et al., 2017). Moreover, while regularization in machine learning generally consists in smoothing over the observation space, in the RL setting, Thodoroff et al. (2018) show that it is possible to smooth over the temporal dimension as well. Furthermore, Zhao et al. (2016) analyze the effects of a regularization using the variance of the policy gradient (the idea is reminiscent of SVRG descent (Johnson & Zhang, 2013)) which proves to provide more consistent policy improvements at the expense of reduced performance. In contrast, as we will see later, AVEC does not change the policy network optimization procedure nor involves any additional computational cost.

Exploration has been studied under different angles in RL, one common strategy is $\epsilon$-greedy, where the agent explores with probability $\epsilon$ by taking a random action. This method, just like entropy regularization, enforces uniform exploration and has achieved recent success in game playing en-

vironments (Mnih et al., 2013; Van Hasselt et al., 2015; Mnih et al., 2016). On the other hand, for most policy-based RL, exploration is a natural component of any algorithm following a stochastic policy, choosing sub-optimal actions with non-zero probability. Furthermore, policy gradient literature contains exploration methods based on uncertainty estimates of values (Kaelbling, 1993; Tokic, 2010), and algorithms which provide intrinsic exploration or curiosity bonus to encourage exploration (Schmidhuber, 2006; Bellemare et al., 2016; Flet-Berliac et al., 2021).

While existing research may share some motivations with our method, no previous work in RL applies the variance of residual errors as an objective loss function. In the context of linear regression, Brown (1947) considers a median-unbiased estimator minimizing the risk with respect to the absolute-deviation loss function (Pham-Gia & Hung, 2001) (similar in spirit to the variance of residual errors), their motivation is nonetheless different to ours. Indeed, they seek to be robust to outliers whereas, when considering noiseless RL problems, one usually seeks to capture those (sometimes rare) signals corresponding to the rewards.

## 3 PRELIMINARIES

### 3.1 BACKGROUND AND NOTATIONS

We consider an infinite-horizon Markov Decision Problem (MDP) with continuous states $s \in \mathcal{S}$, continuous actions $a \in \mathcal{A}$, transition distribution $s_{t+1} \sim \mathcal{P}(s_t, a_t)$ and reward function $r_t \sim \mathcal{R}(s_t, a_t)$. Let $\pi_\theta(a|s)$ denote a stochastic policy with parameter $\theta$, we restrict policies to being Gaussian distributions. In the following, $\pi$ and $\pi_\theta$ denote the same object. The agent repeatedly interacts with the environment by sampling action $a_t \sim \pi(.|s_t)$, receives reward $r_t$ and transitions to a new state $s_{t+1}$. The objective is to maximize the expected sum of discounted rewards:

$$J(\pi) \triangleq \mathbb{E}_{\tau \sim \pi} \left[ \sum_{t=0}^{\infty} \gamma^t r(s_t, a_t) \right], \tag{1}$$

where $\gamma \in [0, 1)$ is a discount factor (Puterman, 1994), and $\tau = (s_0, a_0, r_0, s_1, a_1, r_1, \dots)$ is a trajectory sampled from the environment using policy $\pi$. We denote the value of a state $s$ in the MDP framework while following a policy $\pi$ by $V^\pi(s) \triangleq \mathbb{E}_{\tau \sim \pi} \left[ \sum_{t=0}^{\infty} \gamma^t r(s_t, a_t) | s_0 = s \right]$ and the value of a state-action pair of performing action $a$ in state $s$ and then following policy $\pi$ by $Q^\pi(s, a) \triangleq \mathbb{E}_{\tau \sim \pi} \left[ \sum_{t=0}^{\infty} \gamma^t r(s_t, a_t) | s_0 = s, a_0 = a \right]$. Finally, the advantage function which quantifies how an action $a$ is better than the average action in state $s$ is denoted $A^\pi(s, a) \triangleq Q^\pi(s, a) - V^\pi(s)$.

### 3.2 CRITICS IN DEEP POLICY GRADIENTS

In this section, we consider the case where the value functions are learned using function estimators and then used in an approximation of the gradient. Without loss of generality, we consider the algorithms that approximate the state-value function $V$. The analysis holds for algorithms that approximate the state-action-value function $Q$. Let $f_\phi : \mathcal{S} \to \mathbb{R}$ be an estimator of $\hat{V}^\pi$ with $\phi$ its parameter. $f_\phi$ is traditionally learned through minimizing the mean squared error (MSE) against $\hat{V}^\pi$. At iteration $k$, the critic minimizes:

$$\mathcal{L}_{\text{AC}} = \mathbb{E}_s \left[ \left( f_\phi(s) - \hat{V}^{\pi_{\theta_k}}(s) \right)^2 \right], \tag{2}$$

where the states $s$ are collected under policy $\pi_{\theta_k}$, and $\hat{V}^{\pi_{\theta_k}}(s)$ is an empirical estimate of $V$ (see Section 4.3 for details). Similarly, using $f_\phi : \mathcal{S} \times \mathcal{A} \to \mathbb{R}$ instead, one can fit an empirical target $\hat{Q}^\pi$.

## 4 METHOD: ACTOR WITH VARIANCE ESTIMATED CRITIC

In this section, we introduce AVEC and discuss its correctness, motivations and implementation.

### 4.1 DEFINING AN ALTERNATIVE CRITIC

Recent work (Ilyas et al., 2020) empirically demonstrates that while the value network succeeds in the supervised learning task of fitting $\hat{V}^\pi$ (resp. $\hat{Q}^\pi$), it does not fit $V^\pi$ (resp. $Q^\pi$). We address this

deficiency in the estimation of the critic by introducing an alternative value network loss. Following empirical evidence indicating that the problem is the approximation error and not the estimator *per se*, AVEC adopts a loss that can provide a better approximation error, and yields better estimators of the value function (as will be shown in Section 5.3). At update $k$:

$$\mathcal{L}_{\text{AVEC}} = \mathbb{E}_s\left[\left((f_\phi(s) - \hat{V}^{\pi_{\theta_k}}(s)) - \mathbb{E}_s\left[f_\phi(s) - \hat{V}^{\pi_{\theta_k}}(s)\right]\right)^2\right], \quad (3)$$

with states $s$ collected using $\pi_{\theta_k}$. Note that the gradient flows in $f_\phi$ twice using Eq. 3. Then, we define our bias-corrected estimator: $g_\phi : \mathcal{S} \to \mathbb{R}$ such that $g_\phi(s) = f_\phi(s) + \mathbb{E}_s[\hat{V}^{\pi_{\theta_k}}(s) - f_\phi(s)]$. Analogously to Eq. 3, we define an alternative critic for the estimation of $Q^\pi$ by replacing $\hat{V}^\pi$ by $\hat{Q}^\pi$ and $f_\phi(s)$ by $f_\phi(s, a)$.

**Proposition** (AVEC Policy Gradient). *If $f_\phi : \mathcal{S} \times \mathcal{A} \to \mathbb{R}$ satisfies the parameterization assumption (Sutton et al., 2000) then $g_\phi$ provides an unbiased policy gradient:*

$$\nabla_\theta J(\pi_\theta) = \mathbb{E}_{(s,a)\sim\pi_\theta}\left[\nabla_\theta \log(\pi_\theta(s, a))g_\phi(s, a)\right].$$

*Proof.* See Appendix A. This result also holds for the estimation of $V^{\pi_\theta}$ with $f_\phi : \mathcal{S} \to \mathbb{R}$.

### 4.2 BUILDING MOTIVATION

Here, we present the intuition behind using AVEC for actor-critic algorithms. Tucker et al. (2018) and Ilyas et al. (2020) indicate that the approximation error $\|\hat{V}^\pi - V^\pi\|$ is problematic, suggesting that the variance of the empirical targets $\hat{V}^\pi(s_t)$ is high. Using $\mathcal{L}_{\text{AVEC}}$, our approach reduces the variance term of the MSE (or distance to $V^\pi$) but mechanistically also increases the bias. Our intuition is that since the bias is already quite substantial (Ilyas et al., 2020), it may be possible to reduce the variance enough so that even though the bias increases, the total MSE reduces.

**State-value function estimation.** In this case, optimizing the critic with $\mathcal{L}_{\text{AVEC}}$ can be interpreted as fitting $\hat{V}'^\pi(s) = \hat{V}^\pi(s) - \mathbb{E}_{s'}[\hat{V}^\pi(s')]$ using the MSE. We show that the targets $\hat{V}'^\pi$ are better estimations of $V'^\pi(s) = V^\pi(s) - \mathbb{E}_{s'}[V^\pi(s')]$ than $\hat{V}^\pi$ are of $V^\pi$. To illustrate this, consider T independent random variables $(X_i)_{i\in\{1,...,T\}}$. We denote $X_i' = X_i - \frac{1}{T}\sum_{j=1}^T X_j$ and $\mathbb{V}(X)$ the variance of $X$. Then, $\mathbb{V}(X_i') = \mathbb{V}(X_i) - \frac{2}{T}\mathbb{V}(X_i) + \frac{1}{T^2}\sum_{j=1}^T \mathbb{V}(X_j)$ and $\mathbb{V}(X_i') < \mathbb{V}(X_i)$ as long as $\forall i\ \frac{1}{T}\sum_{j=1}^T \mathbb{V}(X_j) < 2\mathbb{V}(X_i)$, or more generally when state-values are not strongly negatively correlated[1] and not very discordant. This entails that $\hat{V}'^\pi$ has a more compact span, and is consequently easier to fit. This analysis shows that the variance term of the MSE is reduced compared to traditional actor-critic algorithms, but does not guarantee it counterbalances the bias increase. Nevertheless, in practice, the bias is so high that the difference due to learning with AVEC is only marginal and the total MSE decreases. We empirically demonstrate this claim in Section 5.3.

**State-action-value function estimation.** In this case, Eq. 3 translates into replacing $\hat{V}^\pi(s)$ by $\hat{Q}^\pi(s, a)$ and $f_\phi(s)$ by $f_\phi(s, a)$ and the rationale for optimizing the residual variance of the value function instead of the full MSE becomes more straightforward: the practical use of the Q-function is to disentangle the relative values of actions for each state (Sutton et al., 2000). AVEC's effect on relative values is illustrated in a didactic regression with one variable example in Fig. 1 where grey markers are observations and the blue line is our current estimation. Minimizing the MSE, the line is expected to move

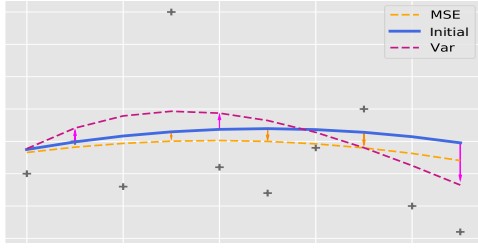

Figure 1: Comparison of simple models derived when $\mathcal{L}_{\text{AVEC}}$ is used instead of the MSE.

towards the orange one in order to reduce errors uniformly. Minimizing the residual variance, it

---
[1] Greensmith et al. (2004) analyze the dependent case: in general, weakly dependent variables tend to concentrate more than independent ones.

is expected to move near the red one. In fact, $\mathcal{L}_{\text{AVEC}}$ tends to further penalize observations that are far away from the mean, implying that AVEC allows a better recovery of the "shape" of the target near extrema. In particular, we see in the figure that the maximum and minimum observation values are quickly identified. Would the approximators be linear and the target state-values independent, the two losses become equivalent since ordinary least squares would provide minimum-variance mean-unbiased estimation.

It should be noted that, as in all the works related to ours, we consider noiseless tasks, *i.e.* the transition matrix is deterministic. As such, there are no outliers and extreme state-action values correspond to learning signals. In this context, high estimation errors indicate where (in the state or action-state space) the training of the value function should be improved.

### 4.3 Implementation

We apply this new formulation to three of the most dominant deep policy gradient methods to study whether it results in a better estimation of the value function. A better estimation of the value function implies better policy improvements. We now describe how AVEC incorporates its residual variance objective into the critics of PPO (Schulman et al., 2017), TRPO (Schulman et al., 2015) and SAC (Haarnoja et al., 2018). Let $\mathcal{B}$ be a batch of transitions. In PPO and TRPO, AVEC modifies the learning of $V_\phi$ (line 12 of Algorithm 1) using:

$$\mathcal{L}^1_{\text{AVEC}}(\phi) = \mathbb{E}_{s \sim \mathcal{B}}\left[\left(f_\phi(s) - \hat{V}^\pi(s)\right) - \mathbb{E}_{s \sim \mathcal{B}}\left[f_\phi(s) - \hat{V}^\pi(s)\right]\right]^2,$$

then $V_\phi = f_\phi(s) + \mathbb{E}_{s \sim \mathcal{B}}[\hat{V}^\pi(s) - f_\phi(s)]$, where $\hat{V}^\pi(s_t) = f_{\phi_{\text{old}}}(s_t) + A_t$ such that $f_{\phi_{\text{old}}}(s_t)$ are the estimates given by the last value function and $A_t$ is the advantage of the policy, *i.e.* the returns minus the expected values ($A_t$ is often estimated using generalized advantage estimation (Schulman et al., 2016). In SAC, AVEC modifies the objective function of $(Q_{\phi_i})_{i=1,2}$ (line 13 of Algorithm 2 in Appendix C) using:

$$\mathcal{L}^2_{\text{AVEC}}(\phi_i) = \mathbb{E}_{(s,a) \sim \mathcal{B}}\left[\left(f_{\phi_i}(s,a) - \hat{Q}^\pi(s,a)\right) - \mathbb{E}_{(s,a) \sim \mathcal{B}}\left[f_{\phi_i}(s,a) - \hat{Q}^\pi(s,a)\right]\right]^2,$$

then $Q_{\phi_i} = f_{\phi_i}(s,a) + \mathbb{E}_{(s,a) \sim \mathcal{B}}[\hat{Q}^\pi(s,a) - f_{\phi_i}(s,a)]$, where $\hat{Q}^\pi(s,a)$ is estimated using temporal difference (see Haarnoja et al. (2018)): $\hat{Q}^\pi(s_t,a_t) = r(s_t,a_t) + \gamma\mathbb{E}_{s_{t+1} \sim \pi}[V_{\bar{\psi}}(s_{t+1})]$ with $\bar{\psi}$ the value function parameter (see Algorithm 2). The reader may have noticed that $\mathcal{L}^1_{\text{AVEC}}$ and $\mathcal{L}^2_{\text{AVEC}}$ slightly differ from Eq. 3. The residual variance of the value function ($\mathcal{L}_{\text{AVEC}}$) is not tractable since *a priori* state-values are dependent and their joint law is unknown. Consequently, in practice, we use the empirical variance proxy assuming independence (*cf.* Appendix D). Greensmith et al. (2004) provide some support for this approximation by showing that weakly dependent variables tend to concentrate more than independent ones. Finally, notice that AVEC does not modify any other part of the considered algorithms whatsoever, which makes its implementation straightforward and keeps the same computational complexity.

## 5 Experimental Study

In this section, we conduct experiments along four orthogonal directions. (a) We validate the superiority of AVEC compared to the traditional actor-critic training. (b) We evaluate AVEC in environments with sparse rewards. (c) We clarify the practical implications of using AVEC by examining the bias in both the empirical and true value function estimations as well as the variance in the empirical gradient. (d) We provide an ablation analysis and study the bias-variance trade-off in the critic by considering two continuous control tasks.

We point out that a comparison to variance-reduction methods is not considered in this paper: Tucker et al. (2018) demonstrated that their implementations diverge from the unbiased methods presented in the respective papers and unveiled that not only do they fail to reduce the variance of the gradient, but that their unbiased versions do not improve performance either. Note that in all experiments we choose the hyperparameters providing the best performance for the considered methods which can only penalize AVEC (*cf.* Appendix E). In all the figures hereafter (except Fig. 3c and 3d), lines are average performances and shaded areas represent one standard deviation.

| Task | SAC | AVEC-SAC | PPO | AVEC-PPO |
|------|-----|----------|-----|----------|
| Ant | 3084 | **3650 ± 127** (**+18%**) | 972 | **1202 ± 148** (**+24%**) |
| AntBullet | 1193 | **2252 ± 82** (**+89%**) | 1174 | **2216 ± 99** (**+89%**) |
| HalfCheetah | 10028 | **11018 ± 102** (**+10%**) | 1068 | **1403 ± 37** (**+31%**) |
| HalfCheetahBullet | 1255 | **1331 ± 184** (**+6%**) | 1329 | **2223 ± 62** (**+67%**) |
| Humanoid | 4084 | **4472 ± 424** (**+10%**) | 391 | **415 ± 4.6** (**+6%**) |
| Reacher | −6.0 | **−5.0 ± 0.1** (**+20%**) | −7.4 | **−5.9 ± 0.3** (**+25%**) |
| Walker2d | 3452 | **4334 ± 128** (**+26%**) | 2193 | **2923 ± 151** (**+33%**) |

Table 1: Average total reward of the last 100 episodes over 6 runs of $10^6$ timesteps. Comparative evaluation of AVEC with SAC and PPO. ± corresponds to a single standard deviation over trials and $(.\%)$ is the change in performance due to AVEC.

## 5.1 CONTINUOUS CONTROL

For ease of comparison with other methods, we evaluate AVEC on the MuJoCo (Todorov et al., 2012) and the PyBullet (Coumans & Bai, 2016) continuous control benchmarks (see Appendix G for details) using OpenAI Gym (Brockman et al., 2016). Note that the PyBullet versions of the locomotion tasks are harder than the MuJoCo equivalents[2]. We choose a representative set of tasks for the experimental evaluation; their action and observation space dimensions are reported in Appendix H. We assess the benefits of AVEC when coupled with the most prominent policy gradient algorithms, currently state-of-the-art methods: PPO (Schulman et al., 2017) and TRPO (Schulman et al., 2015), both on-policy methods, and SAC (Haarnoja et al., 2018), an off-policy maximum entropy deep RL algorithm. We provide the list of hyperparameters and further implementation details in Appendix D and E.

---

**Algorithm 1** AVEC coupled with PPO or TRPO. $J^{\text{ALGO}}$ denotes the policy loss of either algorithm (described in Schulman et al. (2017; 2015)).

---

1: **Input parameters:** $\lambda_\pi \geq 0, \lambda_V \geq 0$
2: **Initialize** policy parameter $\theta$ and value function parameter $\phi$
3: **for** each update step **do**
4:     batch $\mathcal{B} \leftarrow \emptyset$
5:     **for** each environment step **do**
6:         $a_t \sim \pi_\theta(s_t)$
7:         $s_{t+1} \sim \mathcal{P}(s_t, a_t)$
8:         $\mathcal{B} \leftarrow \mathcal{B} \cup \{(s_t, a_t, r_t, s_{t+1})\}$
9:     **end for**
10:     **for** each gradient step **do**
11:         $\theta \leftarrow \theta - \lambda_\pi \hat{\nabla}_\theta J^{\text{ALGO}}(\pi_\theta)$
12:         $\phi \leftarrow \phi - \lambda_V \hat{\nabla}_\phi \mathcal{L}^1_{\text{AVEC}}(\phi)$
13:     **end for**
14: **end for**

---

Table 1 reports the results while Fig. 2 and 8 show the total average return for SAC and PPO. TRPO results are provided in Appendix F for readability. When coupled with SAC and PPO, AVEC brings very significant improvement (on average **+26%** for SAC and **+39%** for PPO) in the performance of the policy gradient algorithms, improvement which is consistent across tasks. As for TRPO, while the improvement in performance is less striking, AVEC still manages to be more efficient in terms of sampling in all tasks. Overall, AVEC improves TRPO, PPO and SAC in terms of performance and efficiency. This does not imply that our method would also improve other policy gradient methods that use the traditional actor-critic framework, but since we evaluate our method coupled with three of the best performing on- and off-policy algorithms, we believe that these experiments are sufficient to prove the relevance of AVEC. Furthermore, in our experiments we do not seek the best hyperparameters for the AVEC variants, we simply adopt the parameters allowing us to optimally reproduce the baselines. Alternatively, if one seeks to evaluate AVEC independently of a considered baseline, further hyperparameter tuning should produce better results. Notice that since no additional calculations are needed in AVEC's implementation, computational complexity remains unchanged.

## 5.2 SPARSE REWARD SIGNALS

Domains with sparse rewards are challenging to solve with uniform exploration as agents receive no feedback on their actions before starting to collect rewards. In such conditions AVEC performs better, suggesting that the *shape* of the value function is better approximated, encouraging exploration.

---

[2]Bullet Physics SDK GitHub Issue.

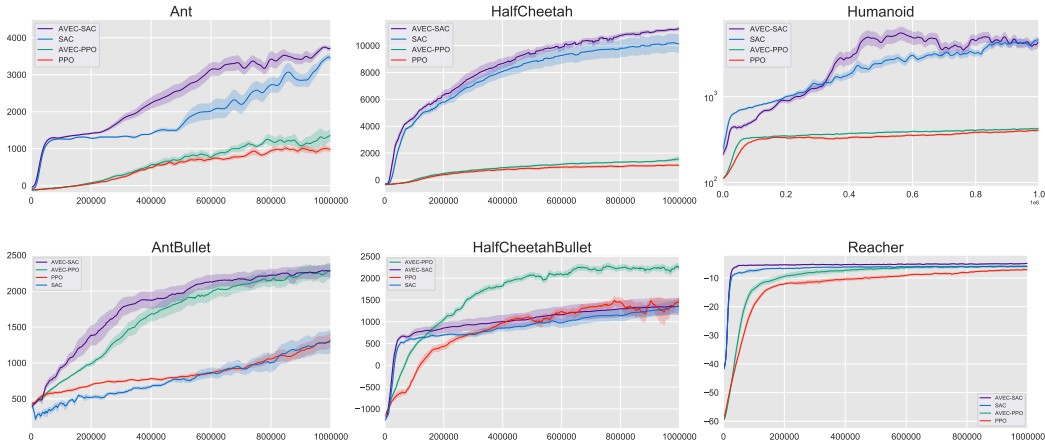

Figure 2: Comparative evaluation (6 seeds) of AVEC with SAC and PPO on PyBullet ("TaskBullet") and MuJoCo ("Task") tasks. X-axis: number of timesteps. Y-axis: average total reward.

The relative value estimate of an unseen state is more accurate: in Section 4.2, AVEC identifies extreme state-values (*e.g.*, non-zero rewards in tasks with sparse rewards) faster. In Fig. 3a and 3b, we report the performance of AVEC in the Acrobot and MountainCar environments: both have sparse rewards. AVEC enhances TRPO and PPO in both experiments. When PPO and AVEC-PPO both reach the best possible performance, AVEC-PPO exhibits better sample efficiency. Fig. 3c and 3d illustrate how the agent improves its exploration strategy in MountainCar: while the PPO agent remains stuck at the bottom of the hill (red), the graph suggest that AVEC-PPO learns the difficult locomotion principles in the absence of rewards and visits a much larger part of the state space (green).

This improved performance in sparse environments can be explained by the fact that AVEC is able to pick up on experienced positive reward more easily. Moreover, the reconstructed shape of the value function is more accurate around such rewarding states, which pushes the agent to explore further around experienced states with high values.

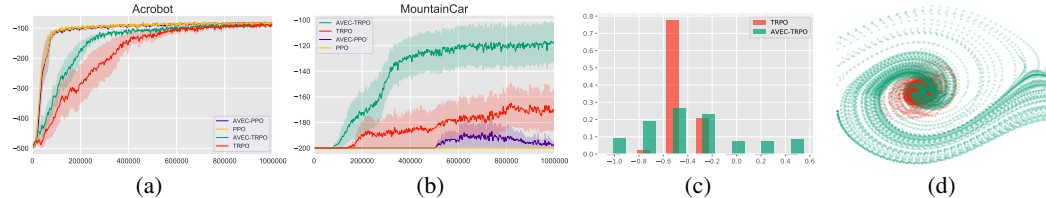

|   (a)   |   (b)   |   (c)   |   (d)   |

Figure 3: (a,b): Comparative evaluation (6 seeds) of AVEC in sparse reward tasks. X-axis: number of timesteps. Y-axis: average total reward. (c,d): Respectively state visitation frequency and phase portrait of visited states of AVEC-TRPO (green) and TRPO (red) in MountainCar.

## 5.3 ANALYSIS OF THE VARIANCE ESTIMATED CRITIC

In order to further validate AVEC, we evaluate the performance of the value network in more detail: we examine (a) the estimation error (distance to the empirical target), (b) the approximation error (distance to the true target) and (c) the empirical variance of the gradient. (a,b) should be put into perspective with the conclusions of Ilyas et al. (2020) where it is found that the critic only fits the empirical value function but not the true one. (c) should be placed in light of Tucker et al. (2018) highlighting a failure of recently proposed state-action-dependent baselines to reduce the variance.

**Learning the Empirical Target.** In Fig. 4, we report the quality of fit (MSE) of the empirical target $\hat{V}^\pi$ in the methods PPO and AVEC-PPO in the AntBullet and HalfCheetahBullet tasks.

We observe that PPO better fits the empirical target than when equipped with AVEC, which is to be expected since vanilla PPO optimizes the MSE directly. This result put aside the remarkable improvement in the performance of AVEC-PPO (Fig. 2) suggests that AVEC might be a better estimator of the true value function. We examine this claim below because if true, it

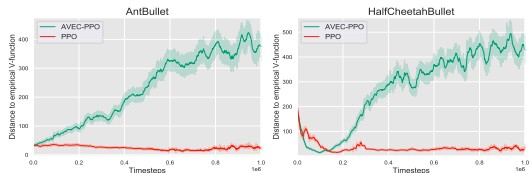

Figure 4: $L_2$ distance to $\hat{V}^\pi$.

would indicate that it is indeed possible to simultaneously improve the performance of the agents and the stability of the method.

**Learning the True Target.** A fundamental premise of policy gradient methods is that optimizing the objective based on an empirical estimation of the value function leads to a better policy. Which is why we investigate the quality of fit of the true target. To approximate the true value function, we fit the returns sampled from the current policy using a large number of transitions $(3 \cdot 10^5)$. Fig. 5 shows that $g_\phi$ is far closer to the true value function half of the time (horizon is $10^6$) than the estimator obtained

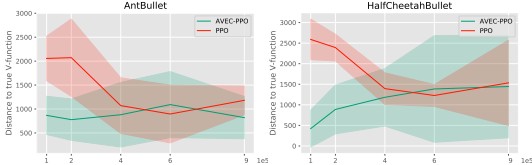

Figure 5: $L_2$ distance to $V^\pi$. X-axis: we run PPO and AVEC-PPO and $\forall t \in \{1, 2, 4, 6, 9\} \cdot 10^5$ we stop training, use the current policy to collect $3 \cdot 10^5$ transitions and estimate $V^\pi$.

with MSE, then as close to it. Comparing Fig. 5 with Fig. 4, we see that the distance to the true target is close to the estimation error for AVEC-PPO, while for PPO, it is at least two orders of magnitude higher at all times. We further investigate these results in Fig. 9 in Appendix B.2 where we study the variation of the squared bias and variance components of the MSE to the true target $(\mathrm{MSE} = \mathrm{Var} + \mathrm{Bias}^2)$. We find, as expected, that using AVEC reduces the variance term significantly while slightly increasing the bias term, which Fig. 5 confirms is negligible since the total MSE is substantially reduced $(\|g_\phi(\mathrm{AVEC}) - V^\pi\|_2 \leq \|V_\phi(\mathrm{PPO}) - V^\pi\|_2)$ where $V_\phi(\mathrm{PPO})$ is the value function estimator in PPO. For completeness, we also analyze the distance to the true target for the Q-function estimator in SAC and AVEC-SAC in AntBullet and HalfCheetahBullet in Appendix B.3, with similar results and interpretation. We conclude that AVEC improves the value function approximation and we expect that the gradient is more stable.

**Empirical Variance Reduction.** We choose to study the gradient variance using the average pairwise cosine similarity metric as it allows a comparison with Ilyas et al. (2020), with which we share the same experimental setup and scales. Fig. 6 shows that AVEC yields a higher average (10 batches per iteration) pairwise cosine similarity, which means closer batch-estimates of

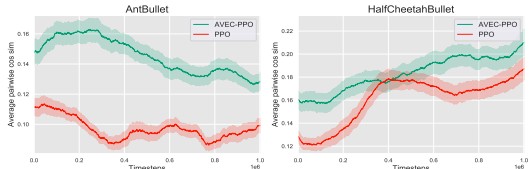

Figure 6: Average gradient cosine-similarity.

the gradient and, in turn, indicates smaller gradient variance. Further analysis with additional tasks is included in Appendix B.4. The variance reduction effect observed in several environments suggests that AVEC is the first method since the introduction of the value function baseline to further reduce the variance of the gradient and improve performance.

### 5.4 ABLATION STUDY

In this section, we examine how changing the relative importance of the bias and the residual variance in the loss of the value network affects learning. For this study, we choose difficult tasks of PyBullet and use PPO because it is more efficient than TRPO and requires less computations than SAC. For an estimator $\hat{y}_n$ of $(y_i)_{i \in \{1, \dots, n\}}$, we write $\mathrm{Bias} = \frac{1}{n} \sum_{i=1}^n (\hat{y}_i - y_i)$ and $\mathrm{Var} = \frac{1}{n-1} \sum_{i=1}^n (\hat{y}_i - y_i - \mathrm{Bias})^2$. Consequently: $\mathrm{MSE} = \mathrm{Var} + \mathrm{Bias}^2$. We denote $\mathcal{L}_\alpha = \mathrm{Var} + \alpha \mathrm{Bias}^2$, with $\alpha \in \mathbb{R}$. In Fig. 7, *Bias-$\alpha$* means that we use $\mathcal{L}_\alpha$ and *Var-$\alpha$* means that we use $\mathcal{L}_{\frac{1}{\alpha}}$. We observe that while no consistent order on the choices of $\alpha$ is identified, AVEC seems to outperform all other weightings. Note that, for readability purposes, the graphs have been split and the curves of AVEC-PPO and PPO are the same in Fig. 7a and 7c, and in Fig. 7b and 7d. A more extensive hyperparameter

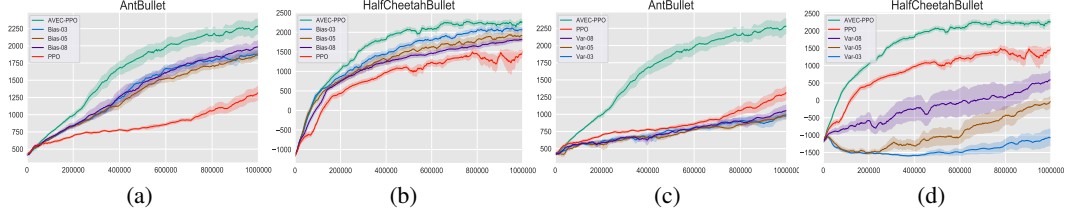

Figure 7: Sensitivity (6 seeds) of AVEC-PPO with respect to (a,b): the bias; (c,d): the variance. X-axis: number of timesteps. Y-axis: average total reward.

study with more $\alpha$ values might provide even higher performances, nevertheless we believe that the stability of an algorithm is crucial for a reliable performance. As such, the tuning of hyperparameters to achieve good results should remain mild.

## 6 DISCUSSION

In this work, we introduce a new training objective for the critic in actor-critic algorithms to better approximate the true value function. In addition to being well-motivated by recent studies on the behaviour of deep policy gradient algorithms, we demonstrate that this modification is both theoretically sound and intuitively supported by the need to improve the approximation error of the critic. The application of Actor with Variance Estimated Critic (AVEC) to state-of-the-art policy gradient methods produces considerable gains in performance (on average +26% for SAC and +39% for PPO) over the standard actor-critic training, without any additional hyperparameter tuning.

First, for SAC-like algorithms where the critic learns a state-action-value function, our results strongly suggest that state-actions with extreme values are identified more quickly. Second, for PPO-like methods where the critic learns the state-values, we show that the variance of the gradient is reduced and empirically demonstrate that this is due to a better approximation of the state-values. In sparse reward environments, the theoretical intuition behind a variance estimated critic is more explicit and is also supported by empirical evidence. In addition to corroborating the results in Ilyas et al. (2020) proving that the value estimator fails to fit $V^\pi$, we propose a method that succeeds in improving both the sample complexity and the stability of prominent actor-critic algorithms. Furthermore, AVEC benefits from its simplicity of implementation since no further assumptions are required (such as horizon awareness Tucker et al. (2018) to remedy the deficiency of existing variance-reduction methods) and the modification of current algorithms represents only a few lines of code.

In this paper, we have demonstrated the benefits of a more thorough analysis of the critic objective in policy gradient methods. Despite our strongly favourable results, we do not claim that the residual variance is the optimal loss for the state-value or the state-action-value functions, and we note that the design of comparably superior estimators for critics in deep policy gradient methods merits further study. In future work, further analysis of the bias-variance trade-off and extension of the results to stochastic environments is anticipated; we consider the problem of noise separation in the latter, as this is the first obstacle to accessing the variance and distinguishing extreme values from outliers.

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

# A  UNBIASED AVEC POLICY GRADIENT

In this section, we consider the case in which the state-action-value function of a policy $\pi_\theta$ is approximated. We prove that given some assumptions on this estimator function, we can use it to yield a valid gradient direction, *i.e.*, we are able to prove policy improvement when following this direction.

In this setting, the critic minimizes the following loss:

$$\mathbb{E}_{(s,a)\sim\pi}\left[(\hat{Q}^{\pi_\theta}(s,a) - f_\phi(s,a) - \mathbb{E}_{(s,a)\sim\pi}[\hat{Q}^{\pi_\theta}(s,a) - f_\phi(s,a)])^2\right].$$

When a local optimum is reached, the gradient of the latter expression is zero:

$$\nabla_\phi \mathcal{L}_{\text{AVEC}} = \mathbb{E}_{(s,a)\sim\pi}\left[(\hat{Q}^{\pi_\theta}(s,a) - f_\phi(s,a) - \mathbb{E}_{(s,a)\sim\pi}[\hat{Q}^{\pi_\theta}(s,a) - f_\phi(s,a)])(\frac{\partial f_\phi(s,a)}{\partial\phi} - \mathbb{E}_{(s,a)\sim\pi}[\frac{\partial f_\phi(s,a)}{\partial\phi}])\right] = 0.$$

In the expression above, the expected value of the partial derivative disappears because the term in the first bracket is centered:

$$\mathbb{E}_{(s,a)\sim\pi}\left[(\hat{Q}^{\pi_\theta}(s,a) - f_\phi(s,a) - \mathbb{E}_{(s,a)\sim\pi}[\hat{Q}^{\pi_\theta}(s,a) - f_\phi(s,a)])\mathbb{E}_{(s,a)\sim\pi}[\frac{\partial f_\phi(s,a)}{\partial\phi}]\right]$$

$$= \mathbb{E}_{(s,a)\sim\pi}\left[\frac{\partial f_\phi(s,a)}{\partial\phi}\right]\mathbb{E}_{(s,a)\sim\pi}[\hat{Q}^{\pi_\theta}(s,a) - f_\phi(s,a) - \mathbb{E}_{(s,a)\sim\pi}[\hat{Q}^{\pi_\theta} - f_\phi]] \qquad = 0$$

$$= 0.$$

Simplifying the gradient at the local optimum becomes:

$$\mathbb{E}_{(s,a)\sim\pi}\left[(\hat{Q}^{\pi_\theta}(s,a) - f_\phi(s,a) - \mathbb{E}_{(s,a)\sim\pi}[\hat{Q}^{\pi_\theta}(s,a) - f_\phi(s,a)])(\frac{\partial f_\phi(s,a)}{\partial\phi})\right] = 0. \tag{4}$$

Then, if we denote $g_\phi = f_\phi(s,a) + \mathbb{E}_{(s,a)\sim\pi}[\hat{Q}^\pi(s,a) - f_\phi(s,a)]$, and use the policy parameterization assumption:

$$\frac{\partial f_\phi(s,a)}{\partial\phi} = \frac{\partial\pi_\theta(s,a)}{\partial\theta}\frac{1}{\pi_\theta(s,a)}, \tag{5}$$

we obtain:

$$\boxed{\nabla_\theta J = \mathbb{E}_{(s,a)\sim\pi_\theta}\left[\nabla_\theta \log(\pi_\theta(s,a))g_\phi(s,a)\right].} \tag{6}$$

*Proof.* By combining the parameterization assumption in Eq. 5 with Eq. 4, we have:

$$\mathbb{E}_{(s,a)\sim\pi_\theta}\left[(\hat{Q}^{\pi_\theta}(s,a) - g_\phi(s,a))\frac{\partial\pi_\theta(s,a)}{\partial\theta}\frac{1}{\pi_\theta(s,a)}\right] = 0. \tag{7}$$

Since the expression above is null, we have the following:

$$\nabla_\theta J = \mathbb{E}_{(s,a)\sim\pi_\theta}[\nabla_\theta \log(\pi_\theta(s,a))\hat{Q}^{\pi_\theta}(s,a)]$$

$$= \mathbb{E}_{(s,a)\sim\pi_\theta}[\nabla_\theta \log(\pi_\theta(s,a))\hat{Q}^{\pi_\theta}(s,a)] - \mathbb{E}_{(s,a)\sim\pi_\theta}[(\hat{Q}^{\pi_\theta}(s,a) - g_\phi(s,a))\frac{\partial\pi_\theta(s,a)}{\partial\theta}\frac{1}{\pi_\theta(s,a)}]$$

$$= \mathbb{E}_{(s,a)\sim\pi_\theta}[\nabla_\theta \log(\pi_\theta(s,a))g_\phi(s,a)].$$

$$\square$$

*Remark.* While the proof seems more or less generic, the assumption in Eq. 5 is extremely constraining to the possible approximators. Sutton et al. (2000) quotes *J. Tsitsiklis* who believes that a linear $g_\phi$ in the features of the policy may be the only feasible solution for this condition.

Concretely, such an assumption cannot hold since neural networks are the standard approximators used in practice. Moreover, empirical analysis (Ilyas et al., 2020) indicates that commonly used algorithms fail to fit the true value function. However, this does not rule out the usefulness of the approach but rather begs for more questioning of the true effect of such biased baselines.

# B ADDITIONAL EXPERIMENTS

## B.1 CONTINUOUS CONTROL: WALKER2D

Fig. 8 shows the total average return for `AVEC` coupled with SAC and PPO on the Walker2d task. Similar to considered other continuous control tasks from MuJoCo and PyBullet, `AVEC` brings a significant performance improvement (+26% for SAC and +33% for PPO), confirming the generality of our approach.

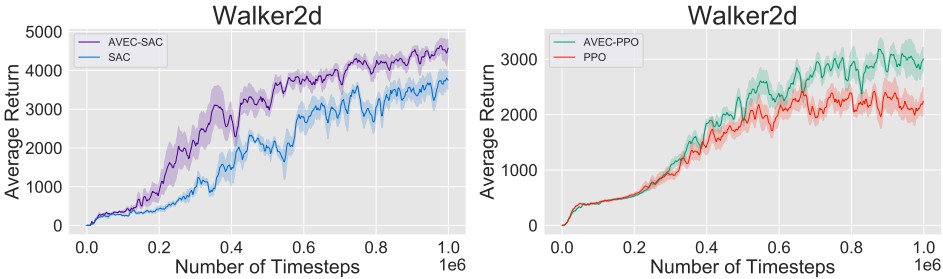

Figure 8: Comparative evaluation (6 seeds) of `AVEC` with SAC (left) and PPO (right) on the Walker2d MuJoCo task. Lines are average performances and shaded areas represent one standard deviation.

## B.2 Variation of the Bias and Variance terms: PPO

In Fig. 9, we show the variation of the bias and variance terms in the MSE between the estimators (of AVEC-PPO and PPO) and the true target: $\mathbb{E}[\|g_\phi - V^\pi\|_2^2] = \text{Bias}(\text{AVEC})^2 + \text{Var}(\text{AVEC})$ and $\mathbb{E}[\|V_\phi(\text{PPO}) - V^\pi\|_2^2] = \text{Bias}(\text{PPO})^2 + \text{Var}(\text{PPO})$ where $V_\phi(\text{PPO})$ is the value function estimator in PPO. We observe that the variance reduction is more substantial than that of the bias. Using those results and Fig. 5 showing that the distance of the estimator to $V^\pi$ is lower when using AVEC confirms that the variance reduction effect counterbalances the bias increase. Note that the % Variation of the Var term is always negative in our experiments, and that the shaded areas that suggest otherwise are merely due to a false assumption of symmetrical deviations, itself due to the assumption of Gaussianity needed to construct confidence intervals.

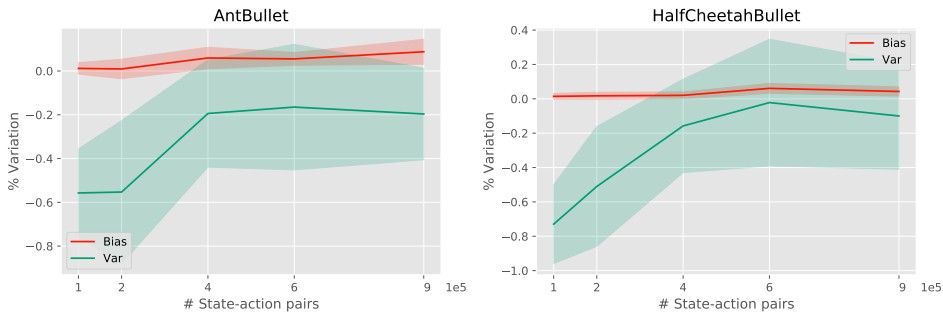

Figure 9: % Variation of the bias and variance terms in the MSE between the estimator and the true target: $\%\text{Variation}(\text{Bias}) = \frac{\text{Bias}^2(\text{AVEC-PPO}) - \text{Bias}^2(\text{PPO})}{\text{Bias}^2(\text{PPO})}$ and $\%\text{Variation}(\text{Var}) = \frac{\text{Var}(\text{AVEC-PPO}) - \text{Var}(\text{PPO})}{\text{Var}(\text{PPO})}$. X-axis: we run PPO and AVEC-PPO and for every $t \in \{1, 2, 4, 6, 9\} \cdot 10^5$, we stop training, use the current policy to interact with the environment for $3 \cdot 10^5$ transitions, and use these transitions to estimate the true value function. Lines are average variations and shaded areas represent one standard deviation (5 seeds).

## B.3 Learning the True Target: SAC

In Fig. 10, we compare the error between the Q-function estimator and the true Q-function for SAC and AVEC-SAC in AntBullet and HalfCheetahBullet. We note a modest but consistent reduction in this error when using AVEC coupled with SAC, echoing the significant performance gains in Fig. 2.

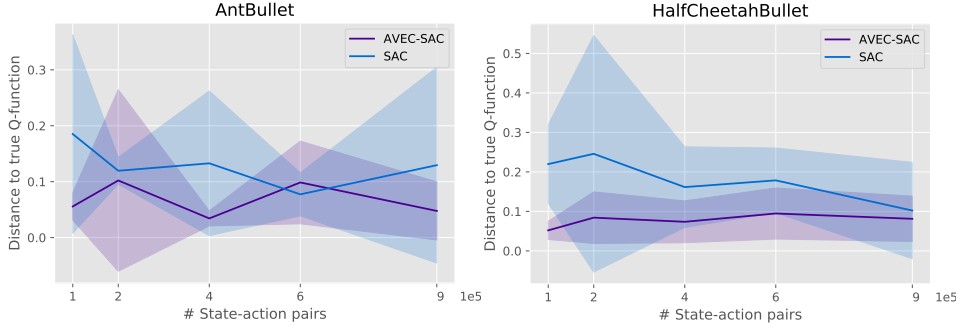

Figure 10: Distance to the true Q-function (SAC). X-axis: we run SAC and AVEC-SAC and for every $t \in \{1, 2, 4, 6, 9\} \cdot 10^5$ we stop training, use the current policy to interact with the environment for $3 \cdot 10^5$ transitions, and use these transitions to estimate the true value function. Lines are average performances and shaded areas represent one standard deviation.

## B.4 Variance Reduction

In Fig. 11, we study the empirical variance of the gradient in measuring the average pairwise cosine similarity (10 gradient measurements) in two additional tasks: HopperBullet and Walker2DBullet. We also vary the trajectory size used in the estimation of the gradient.

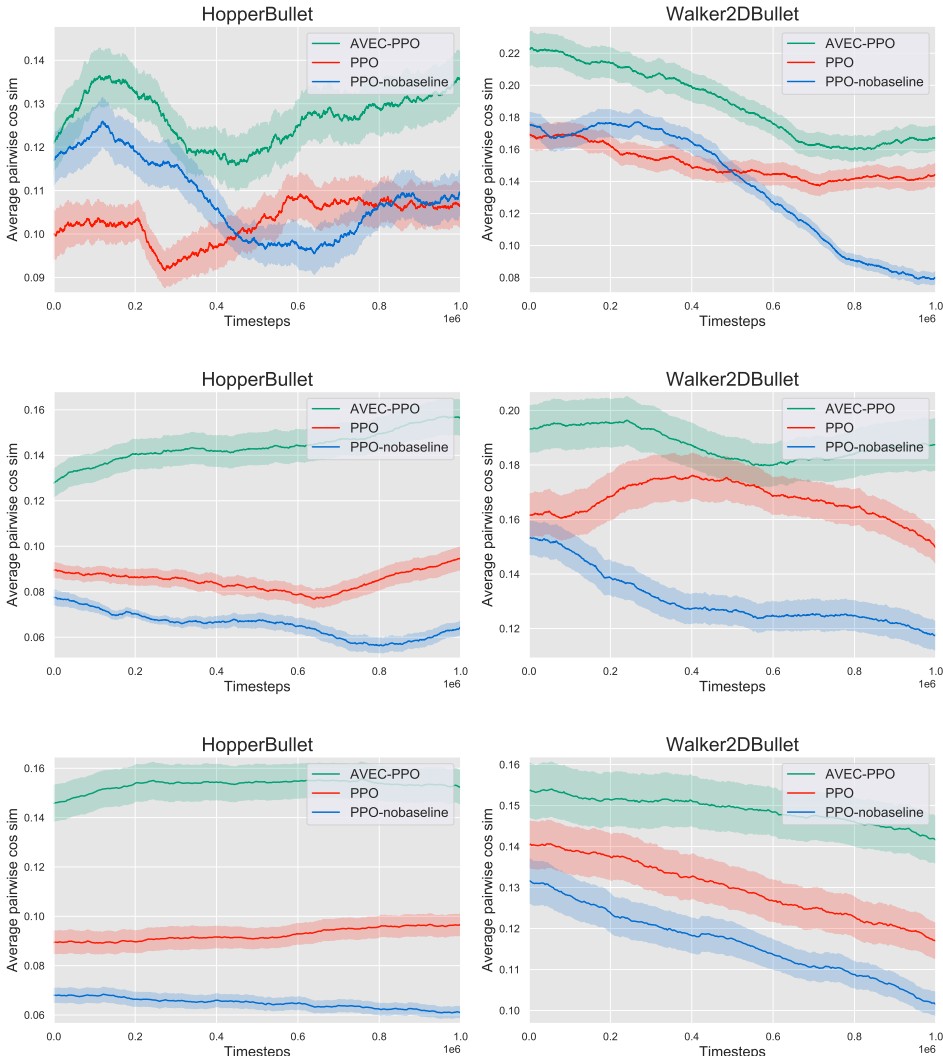

Figure 11: Average cosine similarity between gradient measurements. AVEC empirically reduces the variance compared to PPO or PPO without a baseline (PPO-nobaseline). Trajectory size used in estimation of the gradient variance: 3000 (upper row), 6000 (middle row), 9000 (lower row). Lines are average performances and shaded areas represent one standard deviation.

## C   Implementation of AVEC coupled with SAC

In Algorithm 2, $J_V$ is the squared residual error objective to train the soft value function. See Haarnoja et al. (2018) for further details and notations about SAC, not directly relevant here.

---

**Algorithm 2** AVEC coupled with SAC.

---

1: **Input parameters:** $\beta \in [0, 1], \lambda_V \geq 0, \lambda_Q \geq 0, \lambda_\pi \geq 0$
2: **Initialize** policy parameter $\theta$, value function parameter $\psi$ and $\bar{\psi}$ and Q-functions parameters $\phi_1$ and $\phi_2$
3: $\mathcal{D} \leftarrow \emptyset$
4: **for** each iteration **do**
5:    **for** each step **do**
6:        $a_t \sim \pi_\theta(a_t|s_t)$
7:        $s_{t+1} \sim \mathcal{P}(s_t, a_t)$
8:        $\mathcal{D} \leftarrow \mathcal{D} \cup \{(s_t, a_t, r_t, s_{t+1})\}$
9:    **end for**
10:    **for** each gradient step **do**
11:        sample batch $\mathcal{B}$ from $\mathcal{D}$
12:        $\psi \leftarrow \psi - \lambda_V \hat{\nabla}_\psi J_V(\psi)$
13:        $\phi_i \leftarrow \phi_i - \lambda_Q \hat{\nabla}_{\phi_i} \mathcal{L}^2_{\text{AVEC}}(\phi_i)$ for $i \in \{1, 2\}$
14:        $\theta \leftarrow \theta - \lambda_\pi \hat{\nabla}_\theta J(\pi_\theta)$
15:        $\bar{\psi} \leftarrow \beta\psi + (1 - \beta)\bar{\psi}$
16:    **end for**
17: **end for**

---

## D   Implementation Details

Theoretically, $\mathcal{L}_{\text{AVEC}}$ is defined as the residual variance of the value function (*cf.* Eq. 3). However, state-values for a non-optimal policy are dependent and the variance is not tractable without access to the joint law of state-values. Consequently, to implement AVEC in practice we use the best-known proxy at hand, which is the empirical variance formula assuming independence:

$$\mathcal{L}_{\text{AVEC}} = \frac{1}{T-1} \sum_{t=1}^{T} \left( \left( f_\phi(s_t) - \hat{V}^\pi(s_t) \right) - \frac{1}{T} \sum_{t=1}^{T} \left( f_\phi(s_t) - \hat{V}^\pi(s_t) \right) \right)^2,$$

where $T$ is the size of the sampled trajectory.

# E   EXPERIMENT DETAILS

In all experiments we choose to use the same hyperparameter values for all tasks as the best-performing ones reported in the literature or in their respective open source implementation documentation. We thus ensure the best performance for the conventional actor-critic framework. In other words, since we are interested in evaluating the impact of this new critic, everything else is kept as is. This experimental protocol may not benefit AVEC.

In Table 2, 3 and 4, we report the list of hyperparameters common to all continuous control experiments.

Table 2: Hyperparameters used both in SAC and AVEC-SAC.

| Parameter | Value |
|---|---|
| Adam stepsize | $3 \cdot 10^{-4}$ |
| Discount ($\gamma$) | 0.99 |
| Replay buffer size | $10^6$ |
| Batch size | 256 |
| Nb. hidden layers | 2 |
| Nb. hidden units per layer | 256 |
| Nonlinearity | ReLU |
| Target smoothing coefficient ($\tau$) | 0.01 |
| Target update interval | 1 |
| Gradient steps | 1 |

Table 3: Hyperparameters used both in PPO and AVEC-PPO.

| Parameter | Value |
|---|---|
| Horizon ($T$) | 2048 |
| Adam stepsize | $2.5 \cdot 10^{-4}$ |
| Nb. epochs | 10 |
| Nb. minibatches | 32 |
| Nb. hidden layers | 2 |
| Nb. hidden units per layer | 64 |
| Nonlinearity | tanh |
| Discount ($\gamma$) | 0.99 |
| GAE parameter ($\lambda$) | 0.95 |
| Clipping parameter ($\epsilon$) | 0.2 |

Table 4: Hyperparameters used both in TRPO and AVEC-TRPO.

| Parameter | Value |
|---|---|
| Horizon ($T$) | 2048 |
| Adam stepsize | $1 \cdot 10^{-4}$ |
| Nb. hidden layers | 2 |
| Nb. hidden units per layer | 64 |
| Nonlinearity | tanh |
| Discount ($\gamma$) | 0.99 |
| GAE parameter ($\lambda$) | 0.95 |
| Stepsize KL | 0.01 |
| Nb. iterations for the conjugate gradient | 15 |

# F    COMPARATIVE EVALUATION OF AVEC WITH TRPO

In order to evaluate the performance gains in using AVEC instead of the usual actor-critic framework, we produce some additional experiments with the TRPO (Schulman et al., 2015) algorithm. Fig. 12 shows the learning curves while Table 5 reports the results.

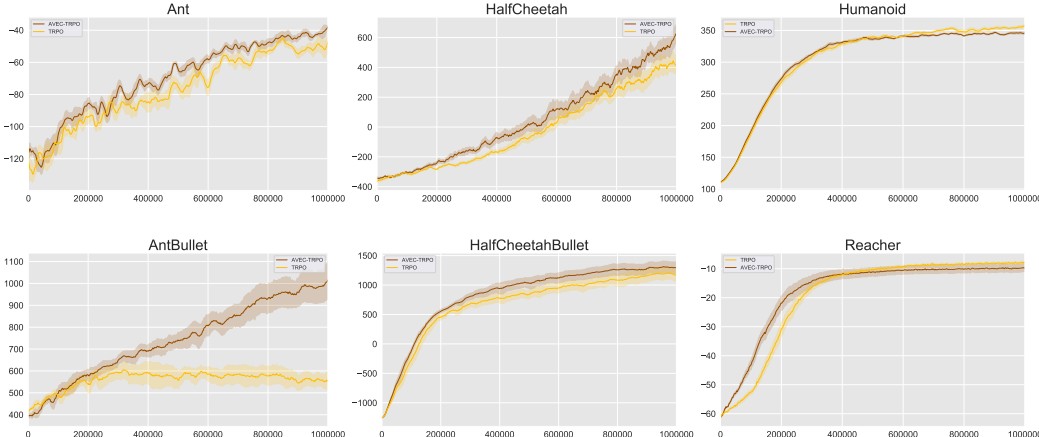

Figure 12: Comparative evaluation of AVEC with TRPO. We run with 6 different seeds: lines are average performances and shaded areas represent one standard deviation.

Table 5: Average total reward of the last 100 episodes over 6 runs of $10^6$ timesteps. Comparative evaluation of AVEC with TRPO. $\pm$ corresponds to a single standard deviation over trials and $(.\%)$ is the change in performance due to AVEC.

| Task | TRPO | AVEC-TRPO |
|---|---|---|
| Ant | $-50.5$ | $\mathbf{-43.5 \pm 2.2}\,(\mathbf{+16\%})$ |
| AntBullet | $564$ | $\mathbf{970 \pm 70}\,(\mathbf{+72\%})$ |
| HCheetah | $346$ | $\mathbf{466 \pm 56}\,(\mathbf{+35\%})$ |
| HCBullet | $1154$ | $\mathbf{1281 \pm 94}\,(\mathbf{+11\%})$ |
| Humanoid | $\mathbf{352}$ | $344 \pm 1.2\,(-3\%)$ |
| Reacher | $\mathbf{-8.5}$ | $-9.9 \pm 1.3\,(-16\%)$ |

# G    ENVIRONMENTS DETAILS

Table 6: Environments details.

| Environment | Description |
|---|---|
| Ant-v2 | Make a four-legged creature walk forward as fast as possible. |
| AntBulletEnv-v0 | Idem.  Ant is heavier, encouraging it to typically have two or more legs on the ground (source: Py-Bullet Guide - url). |
| HalfCheetah-v2 | Make a 2D cheetah robot run. |
| HalfCheetahBulletEnv-v0 | Idem. |
| Humanoid-v2 | Make a three-dimensional bipedal robot walk forward as fast as possible, without falling over. |
| Reacher-v2 | Make a 2D robot reach to a randomly located target. |
| Walker2d-v2 | Make a 2D robot walk forward as fast as possible. |
| Acrobot-v1 | Swing the end of a two-joint acrobot up to a given height. |
| MountainCar-v0 | Get an under powered car to the top of a hill. |

# H    DIMENSIONS OF STUDIED TASKS

Table 7: Actions and observations dimensions.

| Task | $\mathcal{S}$ | $\mathcal{A}$ |
|---|---|---|
| Ant | $\mathbb{R}^{111}$ | $\mathbb{R}^8$ |
| AntBullet | $\mathbb{R}^{28}$ | $\mathbb{R}^8$ |
| HalfCheetah | $\mathbb{R}^{17}$ | $\mathbb{R}^6$ |
| HalfCheetahBullet | $\mathbb{R}^{26}$ | $\mathbb{R}^6$ |
| Humanoid | $\mathbb{R}^{376}$ | $\mathbb{R}^{17}$ |
| Reacher | $\mathbb{R}^{11}$ | $\mathbb{R}^2$ |
| Walker2d | $\mathbb{R}^{17}$ | $\mathbb{R}^6$ |
| Acrobot | $\mathbb{R}^6$ | 3 |
| MountainCar | $\mathbb{R}^2$ | 3 |

