# OpenReview forum: "Learning Value Functions in Deep Policy Gradients using Residual Variance"
_ICLR.cc/2021/Conference — ICLR 2021 Poster_

### Official Review · AnonReviewer1 · 2020-10-19
**New critic loss with good theoretical and empirical motivations**

**Rating:** 8
**Confidence:** 3

**Review:**

This paper presents AVEC, a new critic loss for model-free actor-critic Reinforcement Learning algorithms. The AVEC loss can be used with any actor-critic algorithm, with PPO, TRPO and SAC being evaluated in the paper. The loss builds on the mean-squared-error, and adds a term that minimizes $E_s [f_{\\phi}(s) - \\hat{V}^{\\pi_{\\theta_k}}(s) ]$. The addition of that extra term is motivated by recent research on the stability of actor-critic algorithms, and the benefits obtained by the AVEC loss are empirically demonstrated in numerous environments, with AVEC+PPO, AVEC+SAC and AVEC+TRPO.

Quality: the paper presents an interesting idea, that is simple but well-motivated, and leads to encouraging empirical results. Both the theoretical and empirical motivations are strong.

Clarity: the paper flows well and is quite clear. However, an intuition for what the added term in the AVEC loss is missing. Section 4.2 motivates the added term in a mathematical way, but a few sentences explaining what the added term does, in simple terms, may help the readers understand why AVEC is a better loss than simple MSE.

Originality: the contribution of this paper seems original. It builds on recent work, but the recent work identifies problems while this paper offers an original solution to these problems.

Significance: the fact that AVEC provides good empirical results, and can be used as the critic loss of any actor-critic Reinforcement Learning algorithm, points at the high significance of this work. Many actor-critic implementations can easily be improved by using the AVEC loss. Another positive point is that the paper discusses how to implement the AVEC loss in algorithms that fit a neural network on batches of samples. This really helps implementing the proposed loss, that contains an expectation in an expectation and is therefore not trivial to properly implement.

In general, I like this paper and recommend acceptance.

A few questions/issues:

- An explicit mention of the gradient of the loss, or at least a discussion of where to stop back-propagating gradients, would have been interesting. $f_{\phi}$ appears two times in the AVEC loss, and it is unclear whether the loss contributes to gradients in $f_{\phi}$ two times, or if the expectation over states is first computed (without computing any gradients), and then used as a constant in the rest of the evaluation of the loss.
- As mentioned in "clarity", an intuition of what the added term of the AVEC loss does, especially since it is "inserted" in the mean-squared-error (inside the square), would help the less mathematics-savvy readers. It is not crucial to understand the paper, but the generality of the approach proposed in the paper may lead it to be used often by students, and so an intuition of why AVEC works and what it does would greatly help.

Author response: the authors clarified my questions, so I maintain my recommendation for acceptance.

---

> ### Author Response · Authors · 2020-11-16
> **Response to Reviewer 1**
>
> We would like to thank the reviewer for the positive feedback and thoroughly written review.
>
> Concerning the issues raised by the reviewer:
> * When using AVEC, all the gradients in the total objective remain the same for the coupled method except the gradient of $\mathcal{L}_{\text{AVEC}}$. No stop gradient is used for the value (resp. state-value) function loss: both gradients contribute to the total gradient.
> * Thank you for your comment regarding clarity, we now provide a general intuition at the end of Section 4.2 where we emphasize that this added term is used to focus more on the relative values than on absolute values.

---

> > ### Comment · AnonReviewer1 · 2020-11-18
> > **The paper is now clearer**
> >
> > Thanks for the added details!
> >
> > - This addresses my question. I would just make sure that the fact that the gradient flows in $f_{\phi}$ two times is mentioned just below Equation 3.
> > - The added intuition helps. I don't exactly remember how the paper was before the update, but I appreciate that the current form of the paper also has some intuition about what the contribution does in the abstract. I simply note that from a grammar perspective, and to allow a better understanding from the reader, something is always relative to *something else*, in "learns the relative value of the states (resp. state-action pairs) [relative to each other? to the expectation?] rather than their absolute value".
> > - I'm fully aware that modifying a paper following many comments from many reviewers can be a long and challenging task. I would suggest the authors, just before submitting the final version of the paper, to proof-read everything and check that the paper still flows nicely. For instance, the intuition added in Section 4.2 seems to appear a bit out of nowhere, and could have been put just before Equation 3, or maybe even in the introduction.
> >
> > The mentions that AVEC loss focuses on the relative value of states instead of their absolute value reminds me of the literature on Dueling [1] and Advantage [2] reinforcement learning. I think that citing these papers could further motivate the argument of focusing on relative values, as they both show strong empirical evidence that what matters is the order of Q-Values in a state, not their absolute values.
> >
> > [1]: Wang, Ziyu, et al. "Dueling network architectures for deep reinforcement learning." International conference on machine learning. 2016. (the well-known paper on having a network predict advantage values + state value, instead of Q-Values)
> > [2]: Harmon, Mance E., and Leemon C. Baird III. "Multi-player residual advantage learning with general function approximation." Wright Laboratory, WL/AACF, Wright-Patterson Air Force Base, OH (1996): 45433-7308. (a far lesser-known paper, but much earlier than the other one, that presents a modified Q-operator that focuses on the relative value of actions, and widens the gap between good and bad actions)
> > [3]: Bellemare, Marc G., et al. "Increasing the action gap: New operators for reinforcement learning." arXiv preprint arXiv:1512.04860 (2015). (related to [2] as it considers the gap between Q-Values, but less related to this paper)

---

> > > ### Author Response · Authors · 2020-11-23
> > > **2nd response to Reviewer 1**
> > >
> > > We thank the reviewer for their answers and comments.
> > >
> > > * In the latest version, we now mention that the gradient flows in two times below Equation 3.
> > > * We now write “the value of the states (resp. state-action pairs) *relative to their mean value* rather than the absolute value” in the abstract.
> > > * We agree with the reviewer that the paragraph added in Section 4.2 does seem out of context. We rework it and move it to the Introduction in the newest version.
> > > * Indeed, [1] and [2] are related to this relative value intuition. We add those in the Introduction along with the ranking literature to further introduce the insight.
> > >
> > > In addition to the changes taking effect in the newest revision of the paper, we agree with the reviewer and will take the time at the end of the review process to proof-read and unify the overall flow of the paper.
> > >
> > >
> > > [1] Wang Ziyu, et al. "Dueling network architectures for deep reinforcement learning." International conference on machine learning. 2016.
> > > [2] Harmon Mance E., and Leemon C. Baird III. "Multi-player residual advantage learning with general function approximation." Wright Laboratory, WL/AACF, Wright-Patterson Air Force Base, OH (1996): 45433-7308.

---

### Official Review · AnonReviewer2 · 2020-10-26
**A simple and widely applicable alternative to the squared loss objectives in RL with some evidence of empirical benefits.**

**Rating:** 7
**Confidence:** 4

**Review:**

### Strengths

The paper proposes a simple and elegant idea for changing the value function objectives in deep RL and demonstrates reasonable empirical evidence of it's potential usefulness.  The authors also provide a clearly articulated intuitive motivation and provide experiments to support the proposal.  The idea complements several other algorithms and is therefore quite widely applicable (and easy to try). The analysis of the experiments is also quite interesting and clearly presented.

### Weaknesses

The paper is mostly well written and has interesting theoretical insights as well as empirical analysis. Here are a some weaknesses.

* The theoretical justification for the variance reduction while technically correct, seems like it should be miniscule in theory. For the $T$ independent RV case being analyzed, the condition required for the improvement is that $\Delta  \triangleq 2 \mathbb{V}(X_i) - \frac{1}{T} \sum_{j=1}^T \mathbb{V}(X_j) > 0$, which seems reasonable unless the sample in question is an outlier with a very small variance to begin with. However, the overall reduction itself has another $\frac{1}{T}$ scaling, i.e. the variance reduction over the squared error case is equal to $\frac{\Delta}{T}$, which seems to be vanishingly small as the number of samples $T$ is large even if $\Delta \gg 0$. Note that for the situation where this core idea is being applied, the parameter $T$ is approximately, the number of samples in the expectation over $(s, a)$, which is large in practice.
* The improvements are a good sanity check, but somewhat marginal in many cases (especially given the error bars).

### Additional comments/feedback

* In Section 4.2 paragraph on State-value function estimation line 3, should the targets be $\widehat{V}^\pi$ rather than $V^\pi$?
* In Figure 1, some additional detail on the claims seems necessary (e.g. what parameterization is being considered?)
* In the discussion below the specification for $\mathcal{L}^1_{AVEC}, \mathcal{L}^2_{AVEC}$, the authors say "the reader may have noticed that these equations  slightly differ from Eq. 3", but I am not able to see what difference is being alluded to.
* Figure 4 looks quite surprising in terms of the large qualitative difference between the baseline and AVEC-baseline graphs. Just to be sure, do you measure the fit with respect to $f_\phi$ or the bias corrected version, $g_\phi$? (obviously, the latter makes more sense?).
* The Ablation study in Section 5.4 seems intriguing, but what the conclusions imply seems unclear. It appears the authors were expecting to see some non-zero value of $\alpha$ to improve over $\alpha=0$ (AVEC), but this isn't the case? Some additional clarification here would be useful. Also, it is a bit confusing to separate the plots into two depending on whether the weighting is less than one; as I'm guessing the exact same plot is used for the non-alpha versions in each pair of these graphs?
* In Figure 5, the distance to the true value function seems to be relatively flat (or even mildly increasing) through the entire horizon in both graphs. Is this simply due to the resolution, as I'd expect there to be a drop at least in the initial phase over time.

---

> ### Author Response · Authors · 2020-11-16
> **Response to Reviewer 2**
>
> We thank the reviewer for their time, positive feedback, and insightful comments.
>
> We agree that the variance reduction for a given state value (resp. state-action value) scales with $\frac{1}{T}$. However, we note that to compare AVEC to the squared error case accurately, one should account for the sum of these reductions over visited states in a trajectory, which is equal to $\frac{2T-1}{T} \sum_{j=1}^T \mathbb{V}(X_j)$ and does not scale with $\frac{1}{T}$.
> Furthermore, we emphasize that $T$ is not very large, it represents the size of the trajectories used to approximate the gradient, for example in our experiments it is equal to $2048$.
>
> Concerning the comment of the reviewer on the improvements of our method, we cannot agree with them being marginal: AVEC brings an improvement over the baseline of on average +26% for SAC and +39% for PPO. Moreover, from Table 1, we find that the coefficients of variation (std/mean) are on average 11% for SAC and 9.5% for PPO. Consequently, we believe that empirical improvement conclusions are reasonable.
>
> Thank you for the many additional comments, to which we reply below:
> * It was a typo, we fixed it.
> * The problem depicted in Fig. 1 is a simple example of regression with one variable. We clarify this in Section 4.2.
> * What we would like to highlight in this part of the paper is that the inner empirical expectation is empirically-biased and that it is not possible to propose a bias-corrected version without further restrictive assumptions on the joint law of state-values.
> * We do confirm to the reviewer that Fig. 4 is the L2 distance with respect to the bias-corrected version $g_\phi$. Indeed it is large, which might be surprising at first sight but is not inconsistent with Fig. 5 which shows that PPO’s value estimator is farther from the true target than AVEC-PPO’s from the empirical target.
> * In the ablation study, we question whether there exists a value of $\alpha$ that yields better performance than $\alpha=0$, empirically we find that this is not the case which is why we consider $\mathcal{L}_{\text{AVEC}}$. This is also favorable as it suggests that introducing a weighting with the need to be tuned would not be beneficial. Regarding the separation of the plots, indeed for the same task, AVEC-PPO and PPO are the same curves when the weighting is less than one or not. This was done for readability purposes only. We clarify this in Section 5.4.
> * The flatness of the curves is simply a resolution matter: we decided to plot 5 values ($t \in \{1,2,4,6,9\}.10^5$) corresponding to the order of magnitude chosen in Ilyas et al [1]. mainly because it is computationally demanding and because it suffices in order to compare AVEC to the base algorithm.
>
> [1] Ilyas, Andrew, Logan Engstrom, Shibani Santurkar, Dimitris Tsipras, Firdaus Janoos, Larry Rudolph, and Aleksander Madry. "A Closer Look at Deep Policy Gradients." In International Conference on Learning Representations. 2019.

---

> > ### Comment · AnonReviewer2 · 2020-11-21
> > **Not sure I'm convinced by the correct interpretation of $T$**
> >
> > Thanks for the reply.
> >
> > Regarding the variance reduction comment --  firstly, I'm not sure that the variance reduction should necessarily be a key focus, especially since the objective is being changed and this isn't a case of comparing the variance between two unbiased estimators of the same objective as is typically the case when variance comparisons are invoked. Having said that, I don't fully follow your response. The way I'm understanding it, the variance being referred to is that of the single sample monte carlo estimate for the RHS of Eq (3) (or equivalently, $\mathcal{L}^{1,2} _{AVEC}$).  And $T$ should be the number of samples used to define the Monte carlo estimate in the inner expectation, which is over $(s, a)$ or $s$. I don't quite see how the trajectory length is coming into the picture here.
> >
> > Besides the author comment itself, it seems like the presentation could be improved by more clearly relating the intuition presented at the beginning of 4.2 with what is actually implemented in the algorithm.

---

> > > ### Author Response · Authors · 2020-11-23
> > > **2nd response to Reviewer 2**
> > >
> > > We thank the reviewer for their reply and comments.
> > >
> > > * We agree with the reviewer that the variance reduction for estimators approaching two different targets should not be a key focus. However, we choose to perform this comparison because a proof that the residual variance for an estimator $f_\phi$ obtained using AVEC is less than that of $f_\phi$ obtained using the MSE implies that the new estimator very likely has a reduced residual variance to $V^\pi$. While this does not prove that the distance to the true target $V^\pi$ is improved under AVEC, our intuition is that since the bias is already very high [1], the variance reduction (while collaterally increasing the bias) might be significant enough to reduce the MSE. We further elaborate this point in Section 4.2 and provide empirical evidence in Appendix B.2 of the revised version.
> > > * We would like to recall that, in practice, most actor-critic algorithms (in particular the ones considered in the paper) are trained using, at each update, the empirical targets corresponding to the transitions which were seen during the last few episodes (i.e. batch $\mathcal{B}$ in the paper, composed of the latest 2048 transitions in the case of PPO and TRPO). The critic fits the new targets corresponding to the states visited during those episodes. This is why the $T$ term appearing in $\mathcal{L}_{\text{AVEC}}$ corresponds to the size of $\mathcal{B}$. Note that a batch of transitions can be composed of several episodes: when a terminal state is encountered before $T$ transitions have been collected, the agent is reset to an initial state and continues to perform actions in the environment until the number of transitions in the batch equals $T$.
> > >
> > > [1] Ilyas Andrew, Logan Engstrom, Shibani Santurkar, Dimitris Tsipras, Firdaus Janoos, Larry Rudolph, and Aleksander Madry. "A Closer Look at Deep Policy Gradients." In International Conference on Learning Representations. 2019.

---

### Official Review · AnonReviewer3 · 2020-10-27
**Interesting approach but not sufficient empirical and theoretical evidence to confirm the effectiveness of the approach**

**Rating:** 5
**Confidence:** 3

**Review:**

The paper explores an alternative loss function for fitting critic in Reinforcement Learning. Instead of using the standard mean squared loss between critic predictions and value estimates, the authors propose to use a loss function that also incorporates a variance term. The authors dub the approach AVEC. The authors combine their approach with popular RL algorithms such as SAC and PPO and evaluated on the standard benchmarks for continuous control.

Although the paper demonstrates interesting empirical results, I think that the current experimental evaluation has a number of flaws that prevent me from recommending this paper for acceptance. The paper provides basic motivation but it is lacking thorough theoretical investigation of the phenomena. Also the proposed loss is biased in the stochastic mini batch optimization due to the expectation under the squared term that is not addressed in the paper either. Finally, I have major concerns regarding the experimental evaluation. The set of OpenAI mujoco tasks is different from commonly used tasks in literature. In particular, Hopper and Walker2d, which are used in the vast majority of the literature, are ignored in table 1 and figure 2. This fact raises major concerns regarding generality of the approach.

In conclusion, the paper presents interesting results on some tasks for continuous control. However, the paper requires more thorough experimental evaluation to confirm the statements. Also a deeper theoretical analysis will greatly benefit this work. I strongly encourage the authors to continuous working this approach and revise the paper to improve the theoretical and empirical analysis. This paper presents a very interesting idea but in the current form it is not ready for acceptance.

---

> ### Author Response · Authors · 2020-11-16
> **Response to Reviewer 3**
>
> We thank the reviewer for their time and comments. Below we address the remarks of the reviewer.
>
> First, we consider it important to insist that the main contribution of this paper is to introduce a new loss to learn the critic; we demonstrate the modification does not bias the gradient. Regarding the remark on the bias in the loss, we address this point at the end of Section 4.3 and in Appendix D: “state-values for a non-optimal policy are dependent and the variance is not tractable without access to the joint law of state-values. Consequently, to implement AVEC in practice we use the best-known proxy at hand, which is the empirical variance formula assuming independence“. Moreover, we emphasize that theoretical results in deep policy gradient research are merely a source of motivation as even the most basic assumptions (parameterization assumption for example) are unrealistic: Tucker et al. [1] suggest that gradients resulting from these assumptions are always biased in empirical tasks. Hence the focus of this paper is to provide theoretical insights for both cases: when the critic fits a state value function or a state-action value function. Could the reviewer expand on the additional results considered appropriate? We will do our best to pursue such paths to provide further insights for our method. Nevertheless, we believe this would not belong to this work and would be more appropriate for future investigations.
>
> Concerning the experimental evaluation, we respectfully disagree with the remark of the reviewer. For instance, HalfCheetah and Humanoid are used at least as much commonly in literature and Humanoid is more challenging than Hopper or Walker2d (much larger state and action space). In our experimental protocol we chose a representative set of tasks ranging from moderate (Reacher: $\mathbb{R}^{11} \times \mathbb{R}^{2}$) to very large (Humanoid: $\mathbb{R}^{376} \times \mathbb{R}^{17}$) state and action spaces. Although we are confused by this comment as the reviewer stated earlier that we used “standard benchmarks for continuous control”. Moreover, note that we have already included in Appendix B.2 variance reduction graphs using the (open-source) PyBullet versions of Hopper and Walker2d. Nevertheless, because we sincerely want to address any concern that may come from the reviewer, we include in Table 1 and Appendix B.1 of the revised version of the paper the scores and performance graphs for Walker2d, which compared to Hopper, is more challenging and has a larger state action space ($\mathbb{R}^{17} \times \mathbb{R}^{6}$ vs. $\mathbb{R}^{11} \times \mathbb{R}^{3}$). The results show that using AVEC produces comparable gains in performance: 26% for SAC and 33% for PPO.
>
> [1] Tucker, George, Surya Bhupatiraju, Shixiang Gu, Richard Turner, Zoubin Ghahramani, and Sergey Levine. "The Mirage of Action-Dependent Baselines in Reinforcement Learning." In International Conference on Machine Learning, pp. 5015-5024. 2018.

---

> > ### Comment · AnonReviewer3 · 2020-11-23
> > **Thanks for the update**
> >
> > Thanks for the feedback. Some of my concerns have been addressed and I will raise my score to 5.
> >
> > 1) I agree with the authors that this part might be out of scope of this paper and I will not base my final score on this concern.
> >
> > 2)  I still believe that due to empirical nature of this paper it's extremely important to provide results on this same set of benchmark tasks as prior work. In particular, TD3, SAC, PPO, AWR  and MBPO use Hopper for comparisons with prior work. If Hopper is added, I would raise my score to 6.

---

> > > ### Author Response · Authors · 2020-11-24
> > > **2nd response to Reviewer 3**
> > >
> > > We thank the reviewer for their reply.
> > >
> > > According to your request, we have run the experiments on Hopper and have collected the results for PPO and AVEC-PPO which we add to Appendix B.1. Due to the time limitation, we are not yet able to collect the results for SAC (which is more time-consuming due to the update frequency), but we commit to adding the corresponding graph in the camera-ready version of the paper. Note that with the results with PPO, the average improvement rate goes from 33% to 31%, which remains significant.

---

### Comment · Area_Chair1 · 2020-11-17
**Questions**

1. $\hat{A}, \hat{V}, \hat{Q}$ are defined as "bootstrapped Monte Carlo" estimates. "Bootstrapped" suggests that they are based on an estimator, however, they are in Sec 3.1 before function approximators are introduced. Can the authors elaborate on precisely how these quantities are defined?
2. Sec 4.1 says that the value network fits $\hat{V}$ well but not $V$. This implies that $V$ is not approximated well by $\hat{V}$. Is that what the authors are claiming?
3. Is $g_\theta$ a function? It is an unbiased estimator of $\hat{V}$ which does not approximate $V$ well?
4. What does "consistent" in the proposition mean?
5. Sec 4.2 talks about $\hat{V}'$ defined in terms of expectations, however, the argument talks about subtracting an empirical estimate of the mean that contains $X_i$. How does that affect the argument? Furthermore, while the variance may be reduced, what is the relation between $||\hat{V} - V||$ and $||\hat{V}' - V'||$ as the previous sections argue these are not 0.
6. Sec 4.2 says that the tasks are noiseless. What does this mean? Is the policy deterministic? Does this method apply to stochastic environments?

---

> ### Author Response · Authors · 2020-11-17
> **Response to Area Chair**
>
> We thank the Area Chair for their questions, to which we answer below. We have also uploaded an updated version of the paper.
>
> 1. We agree with the area chair that those quantities were not sufficiently well defined. We give a definition of $\hat{V}$ and $\hat{Q}$ in Section 3.2 and describe in detail in Section 4.3 how the state-value and state-action value functions are estimated in the implementations of PPO, TRPO and SAC.
> 2. According to Ilyas et al. [1], “the value network does succeed in fitting the given loss function [...]. However, the significant drop in performance [referring to the larger distance to the true value function] indicates that the supervised learning problem […] does not lead to [the value network] learning the underlying true value function”. This claim is in accordance with our results in Fig. 4 and 5: in the first half of learning for an agent trained with PPO, $V^\pi$ is not well approximated by $\hat V^\pi $ and AVEC approximates $V^\pi$ better.
> 3. Indeed, $g_\phi$ is a function, we now define it more precisely in Section 4.1. Since translating $f_\phi$ does not increase the loss of the critic, in AVEC we choose $g_\phi$, a bias-corrected version of $f_\phi$ to guarantee that the gradient is unbiased. AVEC leverages the property that $\hat{V'}$ is a better target than $\hat{V}$: it has lower variance. In other words, AVEC modifies the critic’s objective so that $f_\phi$ fits $\hat{V'}$, which approximates $V’$ better than $\hat{V}$ approximates $V$.
> 4. In the paper, we used “consistent” as “unbiased”. We acknowledge this term was confusing and replace it in the revised version.
> 5. Indeed, $\hat{V'}$ is defined in terms of expectations. In the analysis for the independent case, we see a variance reduction effect even when the mean estimate contains $X_i$. An in-depth analysis of the general case requires additional assumptions, we do remark however that the $X_i$ term which appears in the expectation scales with $\frac{1}{T}$ (typically $T=2048$) which means that its influence can be neglected. That being said, our intuition is that since the empirical realizations of $V’$ have lower variance than those of $V$, $\hat{V'}$ is a better estimator of $V’$ than $\hat{V}$ is of $V$, i.e. $\|\|\hat{V'}-V’\|\| \leq \|\|\hat{V}-V\|\|$. Although this does not completely resolve the approximation issue, we demonstrate empirically that it improves the distance to the true value function.
> 6. Noiseless tasks means the transition matrix is deterministic. As such, there are no outliers and extreme state-action values correspond to learning signals. Hence the sentence “high estimation errors indicate where (in the state or action-state space) the training of the value function should be improved”. We clarify this in the corresponding section. The policy is stochastic (defined in Section 3.1). The application of our method in stochastic environments is anticipated in future investigations (cf. end of Section 6).
>
> [1] Ilyas Andrew, Logan Engstrom, Shibani Santurkar, Dimitris Tsipras, Firdaus Janoos, Larry Rudolph, and Aleksander Madry. "A Closer Look at Deep Policy Gradients." In International Conference on Learning Representations. 2019.

---

> > ### Comment · Area_Chair1 · 2020-11-18
> > **RE:**
> >
> > Thank you for the clarifications.
> >
> > 1. Great, clear now.
> > 2. Great, thanks for clarifying.
> > 3. Translating (by a constant) $g_\phi$ does not change the gradient. How does this square with your motivation for defining $g_\phi$. Is lower variance of $\hat{V}'$ the reason for the improvements? Can you provide empirical support for the claim that the targets have substantially lower variance?
> > 4. Great, clear now.
> > 5. I agree that there is an empirical gain, but I do not find the this argument convincing. Is the claim that the bias of the estimator does not change, but the improvement is due to variance reduction? Do these arguments hold for a supervised setting?
> > 6. The policy is stochastic, so the rollouts are still stochastic even with deterministic dynamics. Given that, please elaborate on the claim that "As such, there are no outliers and extreme state-action values correspond to learning signals."

---

> > > ### Author Response · Authors · 2020-11-23
> > > **2nd response to Area Chair**
> > >
> > > We would like to thank the Area Chair for their added questions and remarks. We answer the remaining questions below and upload a revised version of the paper.
> > >
> > >
> > > $3.$ $g_\phi$ is a bias-corrected version of $f_\phi$ to guarantee that the gradient *of the policy* is unbiased, not the gradient of $f_\phi$ (cf. “AVEC Policy Gradient” Proposition). With this respect, translating $f_\phi$ does change the policy gradient. While we give a theoretical intuition behind the improvement suggesting that the targets $V’$ have lower variance, it is unclear how we can additionally confirm this statement empirically as states and state-actions are continuous. Nevertheless, the empirical results added to answer Q5 below do suggest that this is the case: we confirm that AVEC reduces the MSE by reducing the variance term (while collaterally increasing the bias term marginally), which indicates that the variance of $\hat{V}'$ is very likely reduced compared to that of $\hat{V}$.
> > >
> > > $5.$ The claim is that the variance is reduced and the bias is increased, but the variance reduction is far more substantial than the added bias. While it is trivial that $\mathcal{L}_{\text{AVEC}}$ provides an estimator with lower variance, the second part of the claim is not a general mathematical property: the variance reduction is not necessarily enough to counterbalance the bias increase. Thus, this claim does not hold for random supervised learning scenarios*. However in our case, since the bias of the empirical targets is quite substantial (cf. Fig. 5 of this paper and [1,2]), we hypothesized that focusing on relative errors (the variance term of the MSE) could be beneficial. To validate this intuition, we provide tangible evidence by running some additional experiments: in Appendix B.2 we plot the relative change of the bias term and the variance term when using AVEC, we remark very clearly that the variance reduction is far more substantial than that of the bias, and Fig. 5 confirms that it counterbalances the bias increase since the distance to $V^\pi$ is lower when using AVEC.
> > >
> > > $6.$ Indeed, the rollouts are stochastic, and as such high estimation errors indicate the points where the value function should be improved. Put differently, the estimation error contains a variance term due to the randomness of the rollouts, and this variance should be considered as an indication of states to be explored further rather than of possible outliers arising from noise in the reward function or transition matrix.
> > >
> > >
> > > We made the necessary adjustments in the revised version of the paper and would like to thank the Area Chair of their questions - we think this new study further strengthens the overall argumentation for the approach.
> > >
> > >
> > > *To open on your remark concerning supervised settings, we know that in reinforcement learning tasks, the bias of empirical estimators is generally high [1], and as such, we believe that this intuition of focusing on the variance is an interesting research direction that merits additional study, e.g. a time depending loss where the bias term could be increased following annealing strategies based on some indicators of the estimator’s bias might be beneficial.
> > >
> > >
> > > [1] Ilyas Andrew, Logan Engstrom, Shibani Santurkar, Dimitris Tsipras, Firdaus Janoos, Larry Rudolph, and Aleksander Madry. "A Closer Look at Deep Policy Gradients." In International Conference on Learning Representations. 2019.
> > > [2] Tucker George, Surya Bhupatiraju, Shixiang Gu, Richard Turner, Zoubin Ghahramani, and Sergey Levine. "The Mirage of Action-Dependent Baselines in Reinforcement Learning." In International Conference on Machine Learning, pp. 5015-5024. 2018.

---

> > > > ### Comment · Area_Chair1 · 2020-11-23
> > > > **RE:**
> > > >
> > > > 3. To clarify, by translation, I meant adding a constant (that possibly depends on s, but not a). I believe this is also what you meant. This does not change the policy gradient for the same reason that a baseline does not cause bias.
> > > >
> > > > I'm not sure why continuous states and actions mean that variance cannot be measured? It seems that in [2], they quantify variance in similar environments.
> > > >
> > > > 5. If the estimator fits the targets well (as suggested by previous studies), then I would expect the estimator to approximately match the expected value of the targets. So, Fig 5 would suggest that the bias of the AVEC targets is smaller than the standard targets. Can you explain where my understanding is going wrong? Also, in App B.2, can you clarify exactly how the Bias and Var terms are being computed and of what quantities.
> > > >
> > > > 6. With stochastic rollouts, the value function can be perfect and the "estimation error" can be high, so I would be careful about concluding that these are "the points where the value function should be improved." I think the claim about exploration needs further justification if you would like to make such a claim.

---

> > > > > ### Author Response · Authors · 2020-11-24
> > > > > **3rd response to Area Chair**
> > > > >
> > > > > We thank the Area Chair for their interest and additional questions. We answer the remaining questions below.
> > > > > * Indeed, this does not change the policy gradient. Our answer to the initial question of the Area Chair is that the motivation for defining $g_\phi$ is that among the estimators minimizing $\mathcal{L}_{\text{AVEC}}$, it is the one with minimal MSE.
> > > > > * Indeed, in [2] they show that it is possible to evaluate variances, we will study their approach to understand why their estimators are unbiased. If their procedure is suitable we will try and provide similar results to strengthen the motivation of our loss in the camera-ready version.
> > > > > * We do understand there might be some confusion arising from Fig.5. In fact, Fig. 5 presents the L2 distance to $\hat{V}^\pi$, hence a scaled (by the number of samples) MSE, which itself equals $\text{Bias}^2 + \text{Var}$. So Fig. 5 shows the sum of the bias term and the variance term. The intuition is that $\hat{V}’^\pi$ have slightly higher biases than $\hat{V}^\pi$ but significantly lower variances. This leads to $||\hat{V}’^\pi - V^\pi|| \leq ||\hat{V}^\pi - V^\pi||$ even though $\hat{V}’^\pi$ are meant to fit $V’^\pi$. We emphasize that this intuition of a higher bias is purely mathematical and simply due to the fact that we do not minimize the bias as in traditional actor-critic.
> > > > > * **EDIT**: The bias and variance are the measures that come from the decomposition of the distance to the true value function $V^\pi$: $\mathbb{E}[\|\|g_\phi-V^\pi\|\|_2^2]=\text{Bias}(\mathtt{AVEC})^2+\text{Var}(\mathtt{AVEC})$ and $\mathbb{E}[\|\|V_\phi(\mathtt{PPO})-V^\pi\|\|_2^2]=\text{Bias}(\mathtt{PPO})^2+\text{Var}(\mathtt{PPO})$ where $V_\phi(\mathtt{PPO})$ is the value function estimator in PPO. We understand the confusion and make notations clearer in Appendix B.2.
> > > > > * To clarify further, we would like to emphasize that the sentence “the points where the value function should be improved” does not refer to points which we think are optimal but to points that need to be explored to reduce the estimation variability (due to stochastic rollouts) until they are no longer extreme, or forever if those points are truly optimal. We believe that this claim is not counterintuitive because the variability due to rollouts is unavoidable and symmetrical for state-values unless one constructs specific indicators to quantify the estimation variability in each state, which is beyond the scope of this paper.

---

> > > > > > ### Comment · Area_Chair1 · 2020-11-24
> > > > > > **RE:**
> > > > > >
> > > > > > The MSE, Bias, and Var terms are being computed for $g_\phi$ or the targets $\hat{V}$ and $\hat{V}'$? The text suggests it is the former, but the equations suggest the latter.

---

> > > > > > > ### Author Response · Authors · 2020-11-25
> > > > > > > **4th response to Area Chair**
> > > > > > >
> > > > > > > Indeed, the MSE, Bias and Var are computed for $g_\phi$, there was a typo in the equation of the reply. We edited the reply (“**EDIT**: [...]”) and rewrite the correct equations below.
> > > > > > >
> > > > > > > * The bias and variance are the measures that come from the decomposition of the distance to the true value function $V^\pi$: $\mathbb{E}[||g_\phi-V^\pi||_2^2]=\text{Bias}(\mathtt{AVEC})^2+\text{Var}(\mathtt{AVEC})$ and $\mathbb{E}[||V_\phi(\mathtt{PPO})-V^\pi||_2^2]=\text{Bias}(\mathtt{PPO})^2+\text{Var}(\mathtt{PPO})$ where $V_\phi(\mathtt{PPO})$ is the value function estimator in PPO. We also fix the equations in Appendix B.2.

---

### Decision · Program_Chairs · 2021-01-07
**Final Decision**

**Decision:**

Accept (Poster)

**Comment:**

This paper is accepted, however, it could be much stronger by addressing the concerns below.

The theoretical analysis of the proposed methods is weak.
* As far as I can tell, the proposition has more to do with the compatible feature assumption than their method. Furthermore the compatible feature assumption is very strong and not satisfied in any of their experiments.
* Sec 4.2 does not provide strong support for their method. R2 points out issues with their statements about variance and the next subsection argues from an overly simplistic diagram.

The experimental results are promising, however, R3 brought up important issues in the private discussion:
* Their implementation of SAC systematically produces results worse than reported in the original paper (they use a version of SAC with automatically tuned temperature https://arxiv.org/pdf/1812.05905.pdf); 1a) Their SAC gets average returns of 2.5k at 500k steps while the original implementation gets 3k at 500k steps; 1b) Their SAC on HalfCheetah 10k at 1M steps, original paper - 11k at 1M steps; 1c) The same applies to Humanoid, there is no improvement with respect to the original SAC;
* Their approach degrades performance on Hopper.
* They use non-standard hyper parameters for SAC. 0.98 instead of 0.99 for the discount and 0.01 instead of 0.005 for the soft target updates. That might be the main reason why their SAC works worse than the original implementation.
* The authors use the hyper-parameters suggested for HalfCheetahBulletEnv for all continuous control tasks. For HalfCheetah, however, the authors of the stable-baselines repository (which this paper uses) suggest to use the hyper parameters from the original SAC paper (https://github.com/araffin/rl-baselines-zoo/blob/master/hyperparams/sac.yml#L48). Nonetheless, the results for the unmodified SAC reported in this work for HalfCheetah/Hopper/Walker/Ant are subpar to the original results, suggesting that the hyper-parameters for HalfCheetahBulletEnv are suboptimal for these tasks.

Given the simplicity of the change and the promising experimental results (with some caveats), I believe the community will find this paper interesting and will lead to followup work that can patch the theoretical gaps.